# Nitric Oxide (NO) Synthase Inhibitors: Potential Candidates for the Treatment of Anxiety Disorders?

**DOI:** 10.3390/molecules29061411

**Published:** 2024-03-21

**Authors:** Nikolaos Pitsikas

**Affiliations:** Department of Pharmacology, School of Medicine, Faculty of Health Sciences, University of Thessaly, Biopolis, Panepistimiou 3, 415-00 Larissa, Greece; npitsikas@med.uth.gr; Tel.: +30-2410-685-535

**Keywords:** nitric oxide, nitric oxide synthase inhibitors, anxiety

## Abstract

Close to 19% of the world population suffers from anxiety. Current medications for this chronic mental disorder have improved treatment over the last half century or more, but the newer anxiolytics have proved disappointing, and enormous challenges remain. Nitric oxide (NO), an intra- and inter-cellular messenger in the brain, is involved in the pathogenesis of anxiety. In particular, excessive NO production might contribute to its pathology. This implies that it might be useful to reduce nitrergic activity; therefore, molecules aiming to downregulate NO production such as NO synthase inhibitors (NOSIs) might be candidates. Here, it was intended to critically review advances in research on these emerging molecules for the treatment of anxiety disorders. Current assessment indicates that, although NOSIs are implicated in anxiety, their potential anti-anxiety action remains to be established.

## 1. Introduction

Anxiety is a common psychiatric disorder that affects up to 19% of the word population. Anxiety can be considered as an adaptive psychological, physiological, and behavioral situation that makes coping possible when challenged with a real or possible menace [1]. However, anxiety may evolve into a pathological situation and interfere with coping. Anxiety disorders comprise generalized anxiety disorder (GAD), specific and social phobias, post-traumatic stress disorder (PTSD), and panic disorder. In this framework, the conclusions of a conspicuous number of epidemiological studies propose that anxiety disorders have the highest lifetime prevalence estimates (13.6–28.8%) and the earliest age of appearance (11 years) among the various psychiatric diseases [2,3,4].

Up to now, chemicals interfering with the γ-aminobutyric acid (GABA) and serotonergic neurotransmission, like benzodiazepines, partial agonists of the serotonergic 5-HT_1A_ receptor, and selective serotonin reuptake inhibitors (SSRIs) are widely used for the treatment of anxiety disorders. Nonetheless, different types of anxieties do not respond to the above-mentioned therapeutic approaches [5,6]. Moreover, both benzodiazepines and SSRIs are correlated with severe undesired effects, like sedation, cognitive impairments, dependence and withdrawal, sexual disorders, and hyperlipidemia. Additionally, the 5-HT_1A_ receptor partial agonist buspirone, which is well tolerated, is not widely used since it presents important limitations including the slow onset of action and low efficacy [7].

There is a pressing requirement, therefore, to discover and develop new compounds with high efficacy as anti-anxiety-like agents, which should ideally possess a good safety profile [8]. Among the alternative approaches for the treatment of anxiety disorders, the nitrergic system has turned up as a promising target since consistent experimental evidence suggests the nitric oxide (NO) plays a role in anxiety. Thus, molecules targeting NO might be beneficial for the treatment of anxiety disorders. The therapeutic potential of chemicals acting on the nitrergic system like the NO synthase inhibitors (NOSIs) is evaluated in the present review. The PubMed database was utilized for this purpose. In this context, studies written in English and published in peer-reviewed journals were considered.

## 2. Nitric Oxide (NO) 

NO is a soluble, highly diffusible gas with a short half-life (4 s). NO plays an important role in the brain as an intra- and inter-cellular messenger. NO is synthesized by the conversion of L-arginine to L-citrulline by a calcium (Ca^2+^)/calmoduline-dependent enzyme NO synthase (NOS). Three NOS isoforms encoded on different genes have been identified: neuronal NOS (nNOS, NOS type I), which is found in the brain; inducible NOS (iNOS, NOS type II), whose formation is induced by pro-inflammatory agents (cytokines or endotoxin); and endothelial NOS (eNOS, NOS type III), which is localized in the endothelial tissue [9]. A key factor for the synthesis of the NO is the activation of the *n*-methyl-D-aspartate (NMDA) receptor [10]. 

NO exerts its biological effects by interacting with the enzyme-soluble guanylyl cyclase (sGC). Its activation produces cyclic guanosine monophosphate (cGMP) that, in turn, activates a cGMP-dependent protein kinase (PKG) which phosphorylates various proteins [11]. NO action is terminated by the enzyme phosphodiesterase, which neutralizes cGMP [12]. Alternative sGC-independent mechanisms have also been proposed through which NO exerts its biological effects. An important reaction is the *S*-nitrosylation of thiol groups of proteins. Depending on the protein species, *S*-nitrosylation can downregulate or upregulate NO activity. This alternative mechanism responsible for NO’s biological action comprises three cation channels opened by S-nitrosylation, the cyclic nucleotide-gated (CNG) channels; the large conductance Ca^2 +^-activated potassium (BK_Ca_) channels; the ryanodine receptor Ca^2+^ release (RyR) channels; and the enzyme mono (ADP-ribosyl) transferase [13]. Further, NO can react with O_2_ to form N_2_O_3_ that subsequently interacts with the thiol group to produce nitrosothiol, and this process is called nitrosation [13,14]. 

It has also been demonstrated that NO can behave as an internal epigenetic modulator of gene expression and cell phenotype. NO seems to influence key aspects of epigenetic regulation including histone post-translational modifications, DNA methylation, and microRNA levels [15,16].

The involvement of NO in a vast range of physiological processes like cellular immunity [17], vascular tone [18], and neurotransmission [9] is commonly acknowledged. In the central nervous system, it is well documented that NO interferes with synaptic plasticity and cognition [19,20]. Further, NO was found to modulate the release of various neurotransmitters such as acetylcholine, GABA, glutamate, dopamine, and serotonin [19,21,22,23]. Additionally, NO potentiates neuronal survival and differentiation and displays enduring effects on the modulation of transcriptional factors and the action on gene expression [24]. 

The outcome of a series of research reports indicates that abnormally high concentrations of NO are related to anxiety; therefore, chemicals that can downregulate and consequently normalize NO levels might be useful for the treatment of anxiety disorders [25].

## 3. NO and Anxiety

The implication of NO in anxiety has been proposed although its role in this psychiatric disorder is not fully elucidated [25,26]. The conclusion of a genetic study conducted in humans proposes an association of the NOS1 genotype with anxiety [27]. Another clinical study found that serum nitrite concentrations were consistently higher in patients suffering from panic disorder compared to those produced by their control cohorts [28]. 

It has been reported that increments in nNOS and its carboxyl-terminal PDZ ligand (CAPON) complex are associated with anxiogenesis, while disruption of the nNOS-CAPON interaction is related to anxiolysis [29]. The distribution of nNOS neurons in brain structures critically involved in anxiety like the dorsolateral periaqueductal gray (dlPAG) [30], the hypothalamic and amygdaloid nuclei [31], and the hippocampus [32] has been noticed.

It has been also shown that nNOS knockout (KO) mice displayed abnormal anxiety levels with respect to those produced by their WT cohorts [33,34]. The nNOS inhibitor (nNOSI) 7-nitroindazole (7-NI) reduced anxiety-like responses in rats through the downregulation of nitrite levels in the brain [35,36]. In this context, it has been shown that suppression of the nNOS activity in the hippocampus is crucial for the role played by the 5-HT1_A_ serotonergic receptor in anxiolysis [37].

## 4. NOS Inhibitors

NOS inhibitors (NOSIs) are molecules that block the biological action of NO. These compounds present important differences in potency and isoform selectivity. Guanidino-derivatives of L-arginine (e.g., L-NMMA (monomethylarginine), L-N^ω^ nitroarginine, and its methyl ester, L-NOARG and L-NAME) are potent blockers of NOS in vitro and in vivo, but this class of compounds shows poor selectivity towards distinct NOS isoforms [38,39,40]. 

Amino acid derivatives like L-*n*-iminoethylornithine (L-NIO) and L-*N*6-(1-lminoethyl)-lysine (L-NIL) are also able to block NOS. The former has a certain selectivity for the eNOS, and the latter is relatively affine for iNOS. Other NOS inhibitors are substituted citrulline compounds such as thiocitrulline and alkylthiocitrullines (e.g., *S*-methyl and *S*-ethylthiocitrulline, SMTC and SETC). SMTC possesses a certain affinity for nNOS [38,39,40].

Various heterocyclic molecules with variable chemical natures such as 7-NI, L-NPA (N^ω^-propyl-L-arginine), 3-bromo-7-nitroindazole and 2,7-dinitro- indazole], and TRIM [(1-(2-trifluoromethylphenyl) imidazole)] are also powerful nNOS inhibitors. Aminoguanidine (AG), 1400W (3-aminomethylbenzylacetamidine), and certain S-alkylated isothioureas are selective blockers of the iNOS isoform [38,39,40]. 

Non-selective and eNOS inhibitors block eNOS, diminishing endothelial vasodilation and increase blood pressure [41,42,43]. nNOS inhibitors display antinociceptive action [39], and iNOS inhibitors raise blood pressure in septic shock [44]. NOS inhibitors are able to reduce the infarct size in animal models of focal ischemia [45,46,47] and confer neuroprotection as revealed by the preclinical models of Parkinson’s [48,49,50] and Alzheimer’s diseases [51]. The chemical structures of various inhibitors of NOS are illustrated in Figure 1.

## 5. NOS Inhibitors and Anxiety

### 5.1. Effects of Non-Selective NOS Inhibitors in Preclinical Models of Anxiety

The relative literature is summarized in Table 1. Acute peripheral administration of L-NOARG [10 mg/kg, subcutaneously (s.c.)] in mice abolished the anxiolytic effect of the benzodiazepine chlordiazepoxide [52] and antagonized the anxiolytic action of N_2_O [53]. In contrast, the microinjections of L-NOARG (10–100 nmol) and L-NAME (10–200 nmol) induced anxiolysis in the dlPAG of rats [30]. 

L-NAME [10 mg/kg, intraperitoneally, (i.p.)] induced an anti-anxiety-like behavior in rats [54]. A single injection of L-NOARG (30–120 mg/kg) had an anxiogenic effect, while its repeated administration did not affect rats’ performance in behavioral tests evaluating anxiety [55]. Acute systemic challenge with L-NAME (10–60 mg/kg) reduced anxiety, while its chronic application in the same dose range was ineffective [56]. On the contrary, another study carried out in rats obtained opposite results since the acute application of L-NAME (12.5–50 mg/kg, i.p.) induced anxiogenesis [57]. Additionally, L-NAME (50 mg/kg, i.p.) was found to diminish anxiety levels in rats [58]. 

Microinjection of L-NOARG (4 μL) in the hippocampus caused an anxiogenic effect [59]. Systemic administrations of L-NAME (20 and 40 mg/kg) and L-NOARG (20 and 40 mg/kg) in mice produced controversial results since L-NOARG induced anxiolysis while treatment with L-NAME had an anxiogenic effect. Further, both these non-selective NOSIs failed to attenuate anxiety in the stressed mouse [60].

Peripheral acute administration of L-NAME (25–50 mg/kg, i.p.) had an anxiogenic action in rats [61]. Administration of L-NAME (15–300 nmol) into medial amygdala (MeA) and dentate gyrus of hippocampus (DG) along with intra-DG injection of L-NOARG (50–300 nmol) produced an anxiolytic effect [62,63]. Infusion of L-NAME into the dorsal raphe nucleus (DRN), in rats, caused an anxiolytic effect at a low dose (25 nmol), whereas at a higher dose (400 nmol), a sedative effect has been noticed [64]. In agreement with the above results, the microinjection of L-NAME (200 nmol) into the basolateral amygdala (BLA), the lateral septal nucleus (LSN), and the dlPAG induced an anxiolytic effect in rats [65].

Mice that acutely received a low dose of L-NAME (5 mg/kg, i.p.) exhibited an anti-anxiety-like behavior, which potentiated the anxiolytic action of the antidepressant trazodone [66], while those that were treated with a higher dose of L-NAME (50 mg/kg, i.p.) exhibited high anxiety levels [67]. Further, the infusion of L-NAME (50–400 nmol) into rats’ ventral hippocampus (VH) gave rise to an anti-anxiety-like effect [68]. L-NAME delivered at a dose of 10 mg/kg caused anxiolysis [36], while its higher doses (30 and 50 mg/kg) have the opposite effect [69,70]. Finally, the microinjection of L-NAME (10μg/kg) into the medial septum (MS) in rats antagonized the anxiogenic effect of the D_2_/D_3_ dopaminergic agonist quinpirole [71].

### 5.2. Effects of Selective nNOS Inhibitors in Preclinical Models of Anxiety

The relative literature is provided in Table 2. Acute systemic administration of 7-NI in rats (20–120 mg/kg, i.p.) caused an anxiolytic effect [35,72,73], whereas at a higher dose (120 mg/kg, i.p.), a sedative effect was observed [72,73]. Interestingly, the repeated administration of 7-NI (30 mg/kg, i.p.) induced an anti-anxiety-like behavior in rats [35].

In line with the above results, 7-NI (20–120 mg/kg, i.p.) injected in mice reduced anxiety [60,72,73]; however, this effect was confounded by sedation that appeared after its administration in a high dose range (80–120 mg/kg) [72]. Conversely, 7-NI (20 mg/kg, i.p.) induced an anxiogenic effect when injected in stressed mice [60]. 7-NI (10–50 mg/kg, i.p.) and TRIM (10–50 mg/kg, i.p.) induced an anxiolytic effect in mice which, however, was accompanied by sedation and motor incoordination [74].

Rats that received an injection of 7-NI into the MeA (10 nmol) [61], the DG (100 nmol) [63], the DRN (1 nmol) [64], and the VH (20 nmol) [68] displayed an anxiolytic profile. Injection of the L-NPA (0.04, 0.08, and 0.1 nmol) into the dlPAG of rats also caused an anxiolytic effect [75,77,78]. On the contrary, the administration of a higher dose of L-NPA (100 nmol) into the dlPAG of rats had an anxiogenic effect [78]. The joint administration of inactive doses of L-NPA (0.1 nmol) with those of the CB1 cannabinoid agonist amanadine into the dlPAG caused anxiolysis [78]. 7-NI (20–40 mg/kg, i.p.) induced anxiolysis in mice [76], but when administered at 30 mg/kg intraperitoneally, it aggravated the anxiogenic effect of aminophylline [69]. L-NPA (0.04 nmol) injected acutely into the ventromedial prefrontal cortex (vmPFC) of mice prevented the anxiogenic effect of restrained stress [79]. L-NPA (0.4 nmol), like the NMDA receptor antagonist AP7 (1 nmol), infused into the rat bed nucleus of the stria terminalis (BNST) decreased freezing behavior and different autonomic responses (i.e., increase in arterial pressure and heart rate and reduction in tail cutaneous temperature) evidenced in the contextual fear conditioning (CFC) procedure. Further, AP7 downregulated the increase in nitrite levels in conditioned rats [80]. These results along with findings described above [77] propose that CFC expression might be mediated by an NMDA receptor–NO signaling mechanism [80].

### 5.3. Effects of Selective iNOS Inhibitors in Preclinical Models of Anxiety

The relative literature is provided in Table 3. AG (50 mg/kg, i.p.) mitigated anxiety in the stressed male [76,81] and female mice [81], attenuated increase in nitrite concentrations, and counteracted the anxiogenic action of the phosphodiesterase 5 inhibitor sildenafil [76,81]. AG (50 mg/kg, i.p.) administered acutely in mice subjected to the restrained stress potentiated the anxiolytic action of piperine [82].

Chronic treatment with AG (1–20 mg/kg) increased anti-anxiety-like behavior in rats, and it has been demonstrated that this effect was mediated by the activation of the tropomyosin-related kinase B (TRKB) receptor [83]. Either acute (50 mg/kg, i.p.) or repeated (3.75–60 mg/kg, i.p.) treatment with AG alleviated the anxiogenic action of highly refined carbohydrate diet and reduced the abnormal increase in nitrites [84]. Finally, the iNOS inhibitor 1400W injected into the medial PFC (mPFC) induced an anxiolytic effect in rats [85].

In summary, the role exerted by NOSIs in anxiety is very complex and reflects the definition of NO as a Janus molecule or a double-edged sword. Based on the data reported in Table 1, Table 2 and Table 3, NOSIs produce a biphasic action (either anxiolytic or anxiogenic) on anxiety. Their effects seem to be independent of the utilized species (rat or mouse), their gender (male or female), and the used behavioral procedure (e.g., elevated plus maze, elevated T maze, light/dark, open field, holeboard, contextual fear conditioning, social interaction, Vogel, staircase, novelty suppression feeding, and mirror chamber tests). Important differences were observed when the NOSIs effects on anxiety were analyzed considering the treatment schedule (acute vs. repeated administration), route of administration (central or peripheral), and dose range of tested molecules. 

Concerning the treatment schedule, acute treatment with various NOSIs induced a clear biological effect (either anxiolytic or anxiogenic). So far, a small number of studies have addressed the effects of repeated treatment of a NOS inhibitor in anxiety, and the results reported are contradictory. Either anti-anxiety-like behavior [35,83] or ineffectiveness {55,56,73] has been observed following chronic treatment with a NOS inhibitor. The probability that the development of tolerance might cause the failure of treatment cannot be ruled out. Further research is mandatory for elucidating this important issue. 

Regarding the route of administration (central or peripheral), it has been shown that the infusion of NOSIs into specific brain areas, like dlPAG, MeA, VH, BLA, LSN, vmPFC, DRN, DG, and MS, critically involved in anxiety is usually associated with an anxiolytic effect. 

The results of studies in which NOSIs, especially, nsNOSIs and nNOSIs, have been administered peripherally are controversial. It is well documented that the systemic administration of nsNOSIs affects the body and the brain, while treatment with nNOSIs exclusively targets the brain. It seems that the presumed anxiolytic effects of nsNOSIs are associated with their low concentrations, whereas at high doses, these molecules mostly display an anxiogenic-like behavior. It is important to underline that nsNOSIs administered peripherally at high concentrations can cause vasoconstriction, hypertension, and peristaltic dysregulation [41,42,43]. Therefore, it cannot be completely ruled out that the above-described undesired effects could have influenced, to a certain extent, rodents’ performance in the behavioral procedures assessing anxiety. 

The outcome of behavioral studies suggested that systemically injected nNOSIs produced an anxiolytic effect that was revealed in different procedures evaluating anxiety in rodents [35,60,69,72,73,74,76]. However, in some circumstances, and independent of the dose tested, a sedative effect has been noticed following a peripheral treatment with a nNOSI [72,73,74]. 

Finally, iNOSIs have also been found to display an anxiolytic effect in different preclinical behavioral studies (although few), and not many undesired effects have been reported (Table 3). Taking into consideration the above facts, it can be probably concluded that the dose range is perhaps the most critical factor underlying the effects following the systemic administration of NOSIs on anxiety. Small changes in local NO concentrations may be crucial in determining their biological effects [40].

Currently, clinical information dealing with the potential anxiolytic effects of NOSIs is unavailable.

## 6. Potential Mechanism(s) of Action of NOSIs in Anxiety

The exact mechanism(s) of action underlying the anti-anxiety-like effects of NOSIs is still under investigation. Additional research is required to definitively elucidate this important issue. Experimental evidence suggests that anxiety disorders are associated with decreased serotonergic [86], upregulated dopaminergic transmission [87], and low levels of the brain-derived neurotrophic factor (BDNF) [88]. 

It is well documented that stress is implicated in anxiety disorders, and exaggerated nitrergic activity seems to be correlated with stressful stimuli. It has been shown that elevated quantities of NO exert a harmful action on cellular components such as proteins, lipids, and DNA. It seems that NO is a key factor for mediating secondary psychiatric disorders linked to stress including anxiety disorders [26]. Accordingly, the involvement of inflammation and oxidative stress in the pathogenesis of anxiety should be considered. It has been found that various inflammatory markers like cytokines and C-reactive proteins are consistently increased in anxiety disorders [89]. In this context, an increase in reactive oxygen species (ROS) accumulation in neurons and lipid along with protein peroxidation in rodents’ hippocampus and amygdala have been revealed [90]. In line with the above facts, clinical research has shown that different oxidative biomarkers, including nitrites, were detected at high concentrations in patients suffering from social phobia, PTSD [91], or panic [28]. 

NOSIs were found capable of normalizing anxiety-related disturbed serotonergic and dopaminergic transmission. Accordingly, it has been demonstrated that either the local or systemic application of 7-NI increased serotonin levels in the rat VH [92], while the peripheral administration of L-NAME counteracted the anxiogenic action of the D_2_/D_3_ dopaminergic agonist quinpirole in rats [71]. 

In a series of studies, the antioxidant and anti-stress properties of NOSIs have been revealed. L-NAME and 7-NI suppressed the stress-induced enhancement of brain lipid peroxidation activity [93], abolished the oxidative stress caused by methamphetamine [94], and potentiated the antioxidant action of trazodone [66]. AG improved inflammation and oxidative stress biomarkers in brain tissue in a rat model of lipopolysaccharide (LPS)-induced anxiety-like behavior [95]. L-NAME and 7-NI blocked stress-induced c-fos protein expression in the hypothalamic paraventricular nucleus in the rat [96]. Further L-NAME, 7-NI [35,36], and AG [76,81,84] were shown to be capable of mitigating the exaggerated nitrite levels observed in rodents subjected to various anxiety tests. 

Increase in NO production appears to block BDNF synthesis [97], and this inhibitory effect was counteracted by L-NAME and AG [83,98]. It seems that the activation of the TRKB receptor might be the key factor for the panicolytic effect of AG [83]. In Figure 2, the potential mechanisms of action underlying the anti-anxiety-like effects of NOS inhibitors are summarized.

## 7. Conclusions

Overall, contradiction is the appropriate term for defining the preclinical findings reported here. The outcome of the behavioral studies that were carried out for evaluating the anxiolytic-like effects of NOSIs suggested that either anxiolytic or anxiogenic effects were produced by the different NOSIs. It can probably be concluded that some anxiolytic effects were observed mainly in studies in which the acute treatments with NOSIs in a low dose range were tested. Further, the present evaluation detected different limitations in the investigation of the potential anti-anxiety-like effect of NOSIs, including the small number of studies in which a chronic treatment schedule was applied and female rodents were utilized. Regarding the latter issue, it is important to emphasize that anxiety disorders occur in a higher frequency in women than in men [99].

The development of new NOS inhibitors with high efficacy and especially with a robust safety profile might be the target of future experimentations. Future studies should evaluate the potential anxiolytic effects of these novel NOS inhibitors both in male and female rodents. Additionally, treatment strategies should include acute and repeated applications of these new inhibitors. 

In this framework, the biphasic action of NO compounds should be considered. Therefore, small changes in local NO concentration and the time of administration may be key factors in determining their biological effects [40]. Furthermore, in future studies, the effects of a broad dose range of these novel inhibitors on anxiety should be tested.

## Figures and Tables

**Figure 1 molecules-29-01411-f001:**
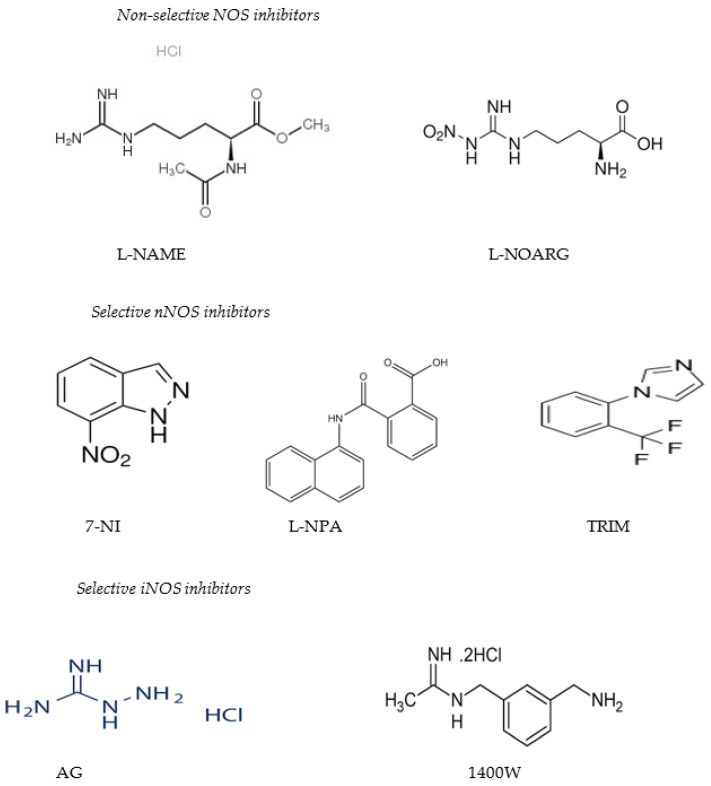
Chemical structures of different inhibitors of the enzyme nitric oxide synthase (NOSIs).

**Figure 2 molecules-29-01411-f002:**
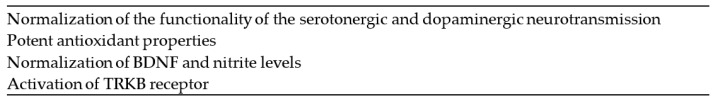
Summary of potential mechanisms of action of NOS inhibitors in anxiety.

**Table 1 molecules-29-01411-t001:** Effects of non-selective nitric oxide synthase inhibitors (nsNOSIs) on animal models of anxiety.

Species	Gender	Drug	Dose Range	Route	Behavioural Task	Effect	Reference
Mouse	Male	L-NOARG	10 mg/kg	s.c. acute	EPM	Reversed chlordiazepoxide-induced anxiolytic effects.	[52]
Mouse	Male	L-NOARG	10 mg/kg	s.c. acute	EPM	Antagonized N_2_O-induced anxiolytic effects.	[53]
Rat	Male	L-NAME L-NOARG	10–200 nmol 10–100 nmol	intra-dlPAG intra-dlPAG	EPM	Anxiolytic effect. Anxiolytic effect.	[30]
Rat	Male	L-NAME	1–20 mg/kg	i.p. acute	EPM	Anxiolytic effect (10 mg/kg).	[54]
Rat	Male	L-NOARG L-NOARG	7.5–120 mg/kg 3.75–60 mg/kg	i.p. acute i.p. chronic	EPM	Anxiogenic (30–120 mg/kg). No effect.	[55]
Rat	Male	L-NAME L-NAME	10–60 mg/kg 15–60 mg/kg	i.p. acute i.p. chronic	EPM	Anxiolytic effect. No effect	[56]
Rat	Male	L-NAME	12.5–50 mg/kg	i.p. acute	EPM SI	Anxiogenic effect. No effect.	[57]
Rat	Male	L-NAME	5, 10, 50 mg/kg	i.p. acute	ETM	Anxiolytic effect (50 mg/kg).	[58]
Rat	Male	L-NOARG	4 μL	intra-hipp.	EPM	Anxiogenic effect.	[59]
Mouse	Male	L-NOARG L-NAME	20, 40 mg/kg 20, 40 mg/kg	i.p. acute i.p. acute	SP model EPM	Anxiolytic effect (20 in control but not in stressed mice. Anxiogenic effect (40 mg/kg) in control, ineffective in stressed mice.	[60]
Mouse	Male	L-NAME	10, 25, 50 mg/kg	i.p. acute	EPM LD HB	Anxiogenic effect (25 and 50 mg/kg) evidenced in LD and HB.	[61]
Rat	Male	L-NAME	50–200 nmol	intra-MeA	EPM	Anxiolytic effect.	[62]
Rat	Male	L-NAME	15–300 nmol	intra-DG	EPM	Anxiolytic effect.	[63]
Rat	Male	L-NOARG L-NAME	50–300 nmol 25, 400 nmol	intra-DRN	Vogel test EPM	Anxiolytic effect (25 nmol) and hypomotility (400 nmol).	[64]
Rat	Male	L-NAME	200 nmol	intra-BLA intra-dlPAG intra-LSN	ETM	Anxiolytic effect (200 nmol). Anxiolytic effect (200 nmol). Anxiolytic effect (200 nmol).	[65]
Mouse	Male	L-NAME	5 mg/kg	i.p. acute i	cFST EPM MC	Anxiolytic effect. Potentiated the antioxidant action of trazodone.	[66]
Mouse	Male	L-NAME	50 mg/kg	i.p. acute	EPM HB OF	Anxiogenic effect. Decreased cGMP.	[67]
Rat	Male	L-NAME	50, 200, 400 nmol	intra-VH	ETM	Anxiolytic effect (200 nmol).	[68]
Rat	Male	L-NAME	10 mg/kg	i.p. acute	RS EPM	Attenuated stress-induced anxiety response.	[36]
Rat	Male	L-NAME	30 mg/kg	i.p. acute	EPM	Aggravated the anxiogenic effect of aminophylline.	[69]
Rat	Male Female	L-NAME	50 mg/kg	i.p acute	RS EPM	Aggravated the anxiogenic action of RS either in male or female rats.	[70]
Rat	Male	L-NAME	10 μg/rat	intra-MS	EPM	Antagonized the anxiogenic effect of the D_2_/D_3_ receptor agonist quinpirole.	[71]

Abbreviations: BLA, basolateral amygdala;cFST, chronic forced swimming test; cGMP, cyclic guanosine monophosphate; DG, dentate gyrus; dlPAG, dorsolateral periaqueductal gray matter; DRN, dorsal raphe nuclei; EPM, elevated plus maze; ETM, elevated T maze; HB, holeboard; hipp., hippocampal; i.p., intraperitoneally; LD, light–dark test; L-NAME, N^ω^-nitro-L-arginine-methylester; L-NOARG, N^G^-nitro-L-arginine; LSN, lateral septal nucleus; MC, mirror chamber; MeA, medial amygdala; MS, medial septum; OF, open field; R.S., restrained stress; s.c., subcutaneously; SI, social interaction; SP, small platform stress test; and VH, ventral hippocampus.

**Table 2 molecules-29-01411-t002:** Effects of neuronal nitric oxide synthase inhibitors (nNOSIs) on animal models of anxiety.

Species	Gender	Drug	Dose Range	Route	Behavioural Task	Effect	Reference
Rat	Male	7-NI	1–80 mg/kg	i.p. acute	EPM SI OF	Anxiolytic effect (20–40 mg/kg). Sedative effect (10 mg/kg).	[72]
Mouse	Male	7-NI	0.1–120 mg/kg	i.p. acute	EPM LD OF	Anxiolytic/sedative effect (80–120 mg/kg).	
Rat	Male	7-NI	3–30 mg/kg 30 mg/kg	i.p. acute i.p. chronic	EPM	Anxiolytic effect (30 mg/kg). Anxiolytic effect (30 mg/kg).	[35]
Rat	Male	7-NI	20–120 mg/kg	i.p. acute	EPM	Anxiolytic effect (90 mg/kg). Sedative effect (120 mg/kg).	[73]
Mouse	Male	7-NI	20–120 mg/kg	i.p. acute	SP EPM	Anxiolytic effect (20–80 mg/kg) in control mice. Anxiogenic effect (20 mg/kg) in stressed mice.	[60]
Mouse	Male	7-NI TRIM	10–50 mg/kg 10–50 mg/kg	i.p. acute i.p. acute	LD OF RR	TRIM (50 mg/kg) but not 7-NI expressed an anxiolytic effect. Both compounds caused sedation and motor incoordination.	[74]
Rat	Male	7-NI	5, 10 nmol	intra-MeA	EPM LD	Anxiolytic effect (10 nmol).	[61]
Rat	Male	7-NI	10–100 nmol	intra-DG	EPM Vogel	Anxiolytic effect (100 nmol).	[62]
Rat	Male	7-NI	1–10 nmol	intra-DRN	EPM	Anxiolytic effect (1 nmol) and hypomotility (10 nmol).	[63]
Rat	Male	L-NPA	0.08 nmol	intra-dlPAG	Vogel	Anxiolytic effect.	[75]
Rat	Male	7-NI	10, 20 nmol	intra-VH	ETM	Anxiolytic effect (20 nmol).	[68]
Mouse	Male	7-NI	20, 40 mg/kg	i.p. acute	stress EPM LD	Anxiolytic effect in unstressed mice.	[76]
Rat	Male	L-NPA	0.04 nmol	intra-dlPAG	CFC	Anxiolytic effect. Attenuated freezing behavior.	[77]
Rat	Male	L-NPA	0.1–100 nmol	intra-dlPAG	EPM	Anxiolytic effect (10 nmol). Anxiogenic effect (100 nmol). At a sub-effective dose (0.1 nmol) in combination with a sub-effective dose (0.1 pmol) of the CB1 cannabinoid receptor agonist anandamide induced an anxiolytic effect.	[78]
Rat	Male	7-NI	30 mg/kg	i.p., acute	EPM	Aggravated the anxiogenic effect of aminophylline.	[69]
Mouse	Male	L-NPA	0.04 nmol	vmPFC	RS EPM	Prevented the anxiogenic effects of restrained stress.	[79]
Rat	Male	L-NPA	0.4 nmol	BNST	CFC	Attenuated freezing behavior.	[80]

Abbreviations: BNST, bed nucleus of the stria terminalis; CFC, contextual fear conditioning; DG, dentate gyrus; dlPAG, dorsolateral periaqueductal grey; DRN, dorsal raphe nuclei; EPM, elevated plus maze; ETM, elevated T maze; hipp., hippocampal; i.p., intraperitoneally; LD, light-dark test; L-NPA, Nω-propyl-L-arginine; MeA, medial amygdala; 7-NI, 7 nitroindazole; OF, open field; R.R., rota rod; R.S., restrained stress; SI, social interaction; SP, small platform stress test; STC, staircase; TRIM,1-(2-trifluoromethylphenyl)imidazole; VH, ventral hippocampus; vmPFC, ventromedial prefrontal cortex.

**Table 3 molecules-29-01411-t003:** Effects of inducible nitric oxide synthase inhibitors (iNOSIs) on animal models of anxiety.

Species	Gender	Drug	Dose Range	Route	Behavioural Task	Effect	Reference
Mouse	Male Female	AG	12.5, 25, 50 mg/kg	i.p. acute	RS EPM LD OF	Anxiolytic effect and normalization of nitrite levels (50 mg/kg) of the stressed but not unstressed mice’ Attenuated the anxiogenic effect of the phosphodiesterase 5 inhibitor sildenafil.	[76,81]
Mouse	Male	AG	50, 100 mg/kg	i.p. acute	RS EPM LD	Anxiolytic effect in stressed mice.	[81]
Mouse	Male	AG	50 mg/kg	i.p. acute	RS EPM LD SI	Potentiated the anxiolytic effect of piperine in stressed mice.	[82]
Rat	Male	AG	1–20 mg/kg	i.p. acute i.p. chronic	ETM	Anxiolytic effect (chronic treatment).	[83]
Mouse	Male	AG	50 mg/kg 3.75–60 mg/kg	i.p. acute	HRCD NSFT	Attenuated the anxiogenic effect and the increased nitrite plasmatic levels caused by HRCD.	[84]
Rat	Male	1400 W	10^−4^, 10^−3^, 10^−2^ nmol	intra-mPFC	RS EPM	Anxiolytic effect in stressed but not unstressed rats.	[85]

Abbreviations: AG, aminoguanidine; EPM, elevated plus maze; ETM, elevated T maze; HRCD, high refined carbohydrate diet; i.p., intraperitoneally; LD, light-dark test; mPFC, medial prefrontal cortex; NSFT, novelty suppression feeding test; OF, open field; R.S., restrained stress; SI, social interaction.

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
