# Peer review of "Nitric Oxide (NO) Synthase Inhibitors: Potential Candidates for the Treatment of Anxiety Disorders?"

_molecules, 2024, doi:10.3390/molecules29061411_

Round 1

Reviewer 1 Report

Comments and Suggestions for Authors

The reviewed manuscript offers valuable insights into the relationship between anxiety and nitric oxide, presenting tables summarizing the impact of nitric oxide synthase inhibitors on animal anxiety models. In general, the manuscript is well-written and provides a review in an organized matter concerning the preclinical assessment of nitric oxide synthase inhibitors on anxiety. I have no specific suggestions for the manuscript except for the following minor suggestion:

Anxiety disorders may be linked to the autonomic nervous system, while glutamate neurotransmission might play a role in the biological mechanisms behind these disorders. Given nitric oxide's involvement in regulating both the autonomic nervous and glutamatergic systems, the authors may consider adding information concerning the possible mechanistic role of glutamate and the autonomic nervous systems in nitric oxide's anxiety regulation.

Author Response

Reviewer 1: comments

General comments

The reviewed manuscript offers valuable insights into the relationship between anxiety and nitric oxide, presenting tables summarizing the impact of nitric oxide synthase inhibitors on animal

anxiety models. In general, the manuscript is well-written and provides a review in an organized matter concerning the preclinical assessment of nitric oxide synthase inhibitors on anxiety. I have no specific suggestions for the manuscript except for the following minor suggestion.

Answer

Author is grateful to Reviewer for his/her comments.

Anxiety disorders may be linked to the autonomic nervous system, while glutamate neurotransmission might play a role in the biological mechanisms behind these disorders. Given nitric oxide's involvement in regulating both the autonomic nervous and glutamatergic systems, the authors may consider adding information concerning the possible mechanistic role of glutamate and the autonomic nervous systems in nitric oxide's anxiety regulation.

Answer

Relative information has been added as kindly requested. Please see lines 214-220.

Reviewer 2 Report

Comments and Suggestions for Authors

After reading the article Nitric Oxide (NO) Synthase Inhibitors: Potential Candidates for the Treatment of Anxiety Disorders?, I found the review carried out by the author very interesting. However, I have several suggestions:

1. Please attach an abstract graphic

2. Please attach how the search for articles for this review was carried out.

3. What information is there at a clinical level?

I suggest checking the following reference 10.22541/au.168494163.33925176/v1

4. Regarding references, the DOI must be attached.

Comments on the Quality of English Language

 Minor editing of English language required

Author Response

Reviewer 2: comments

After reading the article Nitric Oxide (NO) Synthase Inhibitors: Potential Candidates for the Treatment of Anxiety Disorders?, I found the review carried out by the author very interesting. However, I have several suggestions:

Answer

Author is grateful to Reviewer for his/her comments.

  1. Please attach an abstract graphic

Answer

I find interesting Reviewers’ suggestion. Evaluation of the literature did not reach clear conclusions. Results of the different studies conducted aiming to assess the potential anxiolytic profile of NOS inhibitors are conflicting and complex. I believe that the findings described in the manuscript cannot be expressed in a graphical abstract. I hope that a novel table (Table 4) could satisfy Reviewer’s suggestion.

  1. Please attach how the search for articles for this review was carried out.

Answer

Done as kindly requested. Please see lines 44-46.

  1. What information is there at a clinical level?

Answer

Up to now, there is no information if clinical trials have been performed to evaluate the potential anti-anxiety effects of NOS inhibitors. Please see lines 281-282.

I suggest checking the following reference 10.22541/au.168494163.33925176/v1

Answer

Regarding the paper by Azargoonjahromi kindly provided by Reviewer. I found the manuscript interesting. Some issues, however, are not clear. Based on ResearchGate and Authorea this is a preprint/draft (May 2023) which has not been peer reviewed. Data may be preliminary. Further, it is not clear if it is a book chapter or a publication for a journal and its bibliographic reference (if any) is not provided (name of journal, year, volume, pages). I am wondering if I can comment this manuscript.

  1. Regarding references, the DOI must be attached.

Answer

Reporting DOI references number is not included in the Molecules guidelines for the authors. For this reason, I did not include them in the manuscript. Probably, this request is not applicable.

Minor editing of English language required

Answer

An effort to improve language and style of the manuscript has been made.